# A Synthetic Epoxydocosapentaenoic Acid Analogue Ameliorates Cardiac Ischemia/Reperfusion Injury: The Involvement of the Sirtuin 3–NLRP3 Pathway

**DOI:** 10.3390/ijms21155261

**Published:** 2020-07-24

**Authors:** Ahmed M. Darwesh, Wesam Bassiouni, Adeniyi Michael Adebesin, Abdul Sattar Mohammad, John R. Falck, John M. Seubert

**Affiliations:** 1Faculty of Pharmacy and Pharmaceutical Sciences, 2026-M Katz Group Centre for Pharmacy and Health Research, University of Alberta, 11361-97 Ave, Edmonton, AB T6G 2E1, Canada; darweshe@ualberta.ca; 2Department of Pharmacology, Faculty of Medicine and Dentistry, University of Alberta, Edmonton, AB T6G 2E1, Canada; wesam@ualberta.ca; 3Division of Chemistry, Departments of Biochemistry and Pharmacology, University of Texas Southwestern Medical Center, Dallas, TX 75390, USA; adeniyi.adebesin@utsouthwestern.edu (A.M.A.); sattar@utsouthwestern.edu (A.S.M.); J.Falck@utsouthwestern.edu (J.R.F.)

**Keywords:** ischemia-reperfusion injury, EDP surrogates, cardioprotection, mitochondria, Sirtuin 3, NLRP3 inflammasome

## Abstract

While survival rates have markedly improved following cardiac ischemia-reperfusion (IR) injury, the resulting heart damage remains an important issue. Preserving mitochondrial quality and limiting NLRP3 inflammasome activation is an approach to limit IR injury, in which the mitochondrial deacetylase sirtuin 3 (SIRT3) has a role. Recent data demonstrate cytochrome P450 (CYP450)-derived epoxy metabolites, epoxydocosapentaenoic acids (EDPs), of docosahexaenoic acid (DHA), attenuate cardiac IR injury. EDPs undergo rapid removal and inactivation by enzymatic and non-enzymatic processes. The current study hypothesizes that the cardioprotective effects of the synthetic EDP surrogates AS-27, SA-26 and AA-4 against IR injury involve activation of SIRT3. Isolated hearts from wild type (WT) mice were perfused in the Langendorff mode with vehicle, AS-27, SA-26 or AA-4. Improved postischemic functional recovery, maintained cardiac ATP levels, reduced oxidative stress and attenuation of NLRP3 activation were observed in hearts perfused with the analogue SA-26. Assessment of cardiac mitochondria demonstrated SA-26 preserved SIRT3 activity and reduced acetylation of manganese superoxide dismutase (MnSOD) suggesting enhanced antioxidant capacity. Together, these data demonstrate that the cardioprotective effects of the EDP analogue SA-26 against IR injury involve preservation of mitochondrial SIRT3 activity, which attenuates a detrimental innate NLRP3 inflammasome response.

## 1. Introduction

Ischemic heart disease remains one of the primary causes of death worldwide [1,2]. In patients with an ischemic event, early and successful reperfusion or restoration of blood flow to the ischemic myocardium is the most effective treatment to reduce myocardial damage and acute mortality rates [3,4]. However, the benefit of reperfusion is partly attenuated by paradoxical damage to cardiomyocytes that are still viable at the end of the ischemic period, a process known as “ischemia reperfusion” (IR) injury [5,6]. Evidence indicates cardiac dysfunction resulting from IR injury can be linked to mitochondrial dysfunction, excessive oxidative stress and the activation of an inflammatory response [7,8]. Strategies preserving mitochondrial integrity and function have been shown to attenuate IR injury and improve cardiac function [9,10].

Sirtuins (SIRT1-SIRT7) are a class of nicotinamide adenine dinucleotide (NAD^+^)-dependent deacetylase proteins targeted therapeutically for modulating cardiovascular disease (CVD) [11,12]. In the heart, SIRT3 is highly expressed and localized in the mitochondria where it is essential for regulating mitochondrial homeostasis [13,14]. Importantly, more than 65% of mitochondrial proteins are acetylated due to the high pH and the high concentration of acetyl-CoA [15,16]. SIRT3 regulates mitochondrial function by deacetylating, and thereby activating, numerous mitochondrial proteins involved in energy metabolism [17], oxidative stress responses [18,19], mitochondrial dynamics [20] and the electron transport chain (ETC) [21,22]. Studies demonstrate that limiting the loss of SIRT3 following IR injury provides cardioprotective responses and highlight its importance [23,24].

The nucleotide-binding oligomerization domain-like receptor (NLR) family pyrin domain containing 3 (NLRP3) inflammasomes have been shown to play an essential role in the pathogenesis of myocardial IR injury [25,26]. NLRP3 inflammasomes are a large cytosolic platform assembled and activated in response to cellular stress, which can exacerbate inflammatory reactions. During reperfusion, the reactive oxygen species (ROS) burst resulting from damaged mitochondria activates NLRP3 inflammasomes. The oligomerization and assembly of NLRP3 inflammasomes promotes the autocatalytic activation of caspase-1, which in turn exacerbates mitochondrial damage and myocardial injury via pyroptosis [27,28]. Therefore, mitochondrial damage is both a trigger and a target of NLRP3 inflammasome activation [29,30]. Recent evidence reports attempts to preserve SIRT3 activity under stress conditions can block the activation of NLRP3 inflammasome ameliorating the inflammatory insult [31].

Long-chain *n*-3 polyunsaturated fatty acids (*n*-3 PUFA) are essential constituents of the body which have a role in cellular homeostasis and protection against CVD [32,33]. Docosahexaenoic acid (DHA), an abundant *n*-3 PUFA found in mammalian tissues, can be metabolized by cyclooxygenases (COX), lipoxygenases (LOX) and cytochromes P450 (CYP) to a vast range of lipid mediators with different cellular functions [32,34,35]. CYP epoxygenases add oxygen across one of the double bonds of DHA to generate three-membered ethers known as epoxylipids. There are six regioisomeric epoxylipids of DHA termed epoxydocosapentaenoic acids (4,5-, 7,8-, 10,11-, 13,14-, 16,17-, 19,20-EDP) [36]. There is growing evidence indicating that EDPs mediate many of the salutary effects of the parent compound DHA [32,37,38]. Previously, we demonstrated that EDPs can maintain mitochondrial quality and limit NLRP3 inflammasome activation providing a cardioprotective response toward IR injury [38,39,40,41]. However, EDPs are rapidly converted to the corresponding and less biologically active vicinal diol by soluble epoxide hydrolase (sEH) or can be metabolized by other pathways including β-oxidation, chain shortening and chain elongation [42,43]. Accordingly, there is interest in developing methods to enhance their bioavailability [44]. In the current study, we synthesize and investigate the cardioprotective properties of the 19,20-EDP analogue AS-II-75-27 (AS-27) as well as the 16,17-EDP analogues AA-XII-52-4 (AA-4) and SA-II-109-26 (SA-26) against IR injury. The chemical structure of these EDP surrogates possess several key features: (i) a partially saturated carbon backbone to avoid autooxidation and improve physical stability; and (ii) for AS-27, a disubstituted urea that obviates epoxide hydrolysis by sEH and prolongs the half-life (Figure 1) [45,46]. Building upon previous data, we explore the hypothesis that EDP surrogates will attenuate myocardial IR injury through the activation of SIRT3 and subsequent inhibition of NLRP3 inflammasome cascade. 

## 2. Results

### 2.1. SA-26 Improves Post-Ischemic Functional Recovery

Our previous data suggest that EDPs ameliorate cardiac IR injury [40], however, these natural metabolites are rapidly degraded which limit their prolonged use [43]. Therefore, the current study investigates the cardioprotective properties of more chemically and metabolically stable synthetic analogues which were obtained through reducing the number of double bonds and replacing the epoxy-group with a urea group. Notably, preischemic cardiac parameters were similar among all treatment groups. Hearts perfused with SA-26 during the reperfusion period significantly improved postischemic recovery of left ventricular developed pressure (LVDP) compared to vehicle IR group (Figure 2A,B). Consistent with improved postischemic functional recovery, hearts perfused with SA-26 demonstrated better rates of contraction (dP/dt max) and relaxation (dP/dt min) in comparison to the corresponding vehicle IR group (Figure 2C,D). However, perfusion of mouse hearts with either AS-27 or AA-4 did not improve LVDP (Figure 2A,B), rates of contraction (dP/dt max) or relaxation (dP/dt min) (Figure 2C,D) in comparison to the corresponding vehicle IR group. No significant differences were observed in the heart rate at the end of reperfusion between all the study groups (Figure 2E). Interestingly, no differences were observed in postischemic coronary flow rates in any of the treatment groups indicating the cardioprotective effect was not attributable to alterations in hemodynamics in the perfused heart model (Figure 2F). Together, these data suggest that perfusion of hearts with the EDP analogue SA-26 improves post-ischemic functional recovery and ameliorate IR injury.

### 2.2. SIRT3 Function is Preserved in Hearts Perfused with SA-26 

Increased mitochondrial ROS production is implicated in the development of the oxidative damage triggered by IR injury [47]. Over 90% of cellular ROS produced during IR injury is generated by mitochondria as electrons escape the disrupted ETC to combine with molecular O_2_ producing superoxide anions (O^.−2^) [48]. SIRT3 can protect the heart against oxidative stress by directly deacetylating and activating the antioxidant enzyme manganese superoxide dismutase (MnSOD) in mitochondria, significantly enhancing its ability to scavenge ROS [18,49]. MnSOD is the primary mitochondrial antioxidant enzyme that converts O^.−2^to H_2_O_2_, which is further converted to water by catalase [50]. We observed no significant differences in mitochondrial expression of SIRT3 among different groups (Figure 3A). However, SIRT3 activity was significantly decreased in vehicle control hearts subjected to IR injury (Figure 3B). The IR-induced reduction in SIRT3 activity resulted in significantly elevated acetylation and thus decline in the level of activated MnSOD (Figure 3C). Hearts perfused with the analogue SA-26 demonstrated preserved SIRT3 activity and reduced MnSOD acetylation levels compared to the vehicle control IR group (Figure 3B,C). These data suggest perfusion of hearts with SA-26 confers protection against IR-induced oxidative damage by preserving SIRT3 activity accompanied with decreased acetylation and hence maintained MnSOD antioxidant capacity. This was partially supported by assessment of the cardiac levels of malondialdehyde (MDA), the main end product of lipid peroxidation and a key biomarker of oxidative stress [51,52]. MDA cardiac levels were significantly elevated following IR injury (Figure 3D). However, perfusing with SA-26 significantly blunted the accumulation of MDA in post-ischemic hearts (Figure 3D), suggesting less ROS production. In contrast, hearts perfused with the analogues AS-27 and AA-4 neither maintained SIRT3 activity, nor limited the reduction in levels of activated MnSOD nor abrogated the accumulation of MDA compared to the vehicle control IR group. 

Sirtuins are a family of NAD^+^-dependent histone deacetylases. Therefore, the rate-limiting factors of SIRT3 activity are the availability of the co-factor NAD^+^ and the ratio of oxidized to the reduced NAD^+^ [53,54]. We measured NAD^+^ and NADH levels and their relative ratio in the setting of IR injury. There were no significant differences in the total NAD and NADH content between any of the study groups (Figure 3E,F). However, NAD^+^ content and NAD^+^/NADH ratio were markedly lower in hearts subjected to IR injury compared to aerobic CT (Figure 3F,G). Interestingly, perfusion of mice hearts with the analogue SA-26 maintained NAD^+^ levels and thus restored NAD^+^/NADH ratio compared to the vehicle IR (Figure 3G,H). On the other hand, hearts perfused with the EDP analogues AS-27 or AA-4 did not restore the NAD^+^/NADH ratio compared to the vehicle IR group (Figure 3G,H). These findings could explain the observed reduction in SIRT3 activity and the impaired post-ischemic recovery in the hearts perfused with these analogues.

Collectively, these data provide evidence that SA-26 prevents cardiac mitochondrial dysfunction and protects the heart against IR-induced oxidative injury via maintaining SIRT3 activity and consequently activating the major mitochondrial antioxidant enzyme MnSOD.

### 2.3. SA-26 Mitigates Mitochondrial Damage in Response to IR Injury

Accumulating literature revealed that cardiac dysfunction resulting from IR injury can be directly linked to mitochondrial damage [9,10]. We investigated the changes in the protein levels of the ETC components succinate dehydrogenase subunit A (SDH-A, Complex II) and cytochrome c oxidase subunit IV (COX IV), the terminal enzyme of the mitochondrial respiratory chain (Complex IV). Immunoblotting results revealed no significant differences in the protein expression of SDH-A or COX IV among different treatment groups (Figure 4A,B). Additionally, there were no significant changes in the expression level of citrate synthase (CS) suggesting differences were not attributable to changes in mitochondrial biomass (Figure 4C). 

Mitochondrial dysfunction resulting from IR injury could stem from aberrations in proteins regulating mitochondrial dynamics, particularly dynamin-related protein 1 (DRP1) and optic atrophy 1 (OPA1) [55]. An increased mitochondrial localization of the active DRP1 following marked mitochondrial damage results in excessive mitochondrial fission [56,57]. OPA1, a protein localized in the inner mitochondrial membrane, is considered an important determinant of mitochondrial integrity, fusion and function [58,59]. Loss of OPA1 impairs mitochondrial fusion, perturbs cristae structure, and increases the susceptibility of cell death [20,60]. In the current study, we observed a marked increase in mitochondrial expression of DRP1 and a significant decrease in mitochondrial OPA1 levels in the vehicle IR group compared to the aerobic CT (Figure 4D,E). However, perfusion of hearts with SA-26 limited the mitochondrial localization of DRP1 and the loss of OPA1 protein compared to the IR group (Figure 4D,E), which supports the notion of reduced mitochondrial injury. However, neither AS-27 nor AA-4 prevented the mitochondrial localization of DRP1 compared to the vehicle IR group (Figure 4D,E). Notably, the level of OPA1 protein was not significantly reduced in hearts perfused with either AS-27, SA-26 or AA-4 compared to the aerobic CT. However, only hearts perfused with the EDP analogue SA-26 maintained OPA1 protein at a significantly higher level compared to the IR group. 

To further assess the effect of these changes on the mitochondrial function, we evaluated cardiac ATP levels in the different study groups. Cardiac ATP levels were significantly lower in the vehicle IR group as well as hearts perfused with the analogues AS-27 or AA-4 compared to the aerobic group. Interestingly, hearts perfused with the analogue SA-26 maintained ATP content at a significantly higher level than the vehicle IR group (Figure 4F). Altogether, these findings demonstrate that SA-26 could improve cardiac recovery following IR injury via ameliorating mitochondrial dysfunction and consequently maintaining adequate supply of cellular energy.

### 2.4. SA-26 Inhibits IR-Induced NLRP3 Assembly on the Mitochondrial Membrane 

Growing evidence demonstrates mitochondrial damage can be both a trigger and a target of NLRP3 inflammasome cascade [30,61]. Briefly, assembly of the NLRP3 inflammasome complex requires recruitment of NLRP3 and pro-caspase 1 to mitochondria in the perinuclear region, leading to increased co-localization of NLRP3 on mitochondrial membrane [62,63,64]. Afterwards, the activated NLRP3 inflammasome induces cleavage of pro-caspase-1 to generate the active caspase 1 [30]. To determine the ability of different analogues to interfere with the NLRP3 inflammasome cascade activation in the current experimental model, expression levels of NLRP3 and cleaved caspase 1 proteins were determined in the mitochondrial fractions and caspase-1 activity was assessed in cytosolic fractions. As shown in Figure 5, immunoblot analysis revealed hearts subjected to IR injury or even perfused with the analogue AS-27 showed significantly higher protein levels of both NLRP3 and caspase-1 in mitochondrial fractions (Figure 5A,C) and demonstrated higher caspase-1 activity compared to aerobic control (Figure 5B). Perfusion with SA-26 significantly prevented the IR-induced upregulation of NLRP3 protein and caspase-1 in mitochondrial fractions (Figure 5A,C). Moreover, active caspase-1 levels were the same between aerobic controls and hearts perfused with SA-26 (Figure 5B). Perfusion of mice hearts with the analogue AA-4 failed to attenuate postischemic NLRP3 or caspase-1 activation compared to the vehicle IR group (Figure 3A–C). Cumulatively, these data demonstrate that the analogue SA-26 limits the activation of NLRP3 inflammasomes in response to IR injury.

## 3. Discussion

In the current study, we provide evidence for a novel cardioprotective mechanism of the newly synthetic EDP surrogate SA-26. Our results suggest that the perfusion of hearts with SA-26 during the reperfusion period ameliorates IR injury by maintaining mitochondrial integrity and function. Hearts perfused with SA-26 showed improved SIRT3 activity, enhanced antioxidant capacity and limited oxidative injury. This effect was associated with the inhibition of the assembly and mitochondrial translocation of NLRP3 inflammasomes. To the best of our knowledge, this is the first study reporting the involvement of SIRT3 in mediating the cardioprotective properties of the CYP-derived epoxylipids of *n*-3 PUFAs or their analogues against IR injury (Figure 6). 

Emerging evidence has identified EDPs, a group of epoxylipids generated endogenously through the CYP-metabolism of the *n*-3 PUFA DHA, as potent lipid mediators with better beneficial effects than its parent compound, including protection against CVDs [32,37,65]. Although the precise mechanism(s) remains elusive, our group and others have demonstrated that pathways targeting the mitochondria and NLRP3 inflammasome activation are involved [38,40,66]. As such, elevation of intracellular EDPs has been hypothesized to impart cardioprotective effects against several CVDs. However, EDPs are considered chemically and metabolically labile mediators which limit their use as therapeutic or pharmacological agents [36]. For instance, the peak plasma concentration of these metabolites occurred 1 min post intraperitoneal injection, indicating a relatively short half-life for these metabolites [67,68]. Moreover, EDPs may slowly darken and polymerize upon exposure to light, heat or oxygen because of the presence of their multiple double bonds [44]. Considerable interest has arisen in developing methods or surrogates to enhance the bioavailability of EDPs, particularly 16,17-EDP and 19,20-EDP, the two most abundant and potent regioisomers [69,70]. One of the approaches to enhance the bioavailability and consequently the beneficial effects of these metabolites would be combining EDPs with an sEH inhibitor (sEHi) to limit their degradation. For instance, Capozzi et al. demonstrated the exogenous addition of 19,20-EDP combined with the sEHi 12-(3-adamantane-1-yl-ureido)-dodecanoic acid (AUDA) in an inflammation model of tumor necrosis factor alpha (TNFα)-stimulated human retinal microvascular endothelial cells, significantly inhibited the expression of the inflammatory markers vascular cell adhesion molecule 1 (VCAM-1) and intercellular adhesion molecule-1 (ICAM-1). In contrast, the diol metabolites of 19,20-EDP hydrolysis, 19,20-dihydroxydocosapentaenoic acid (19,20-DHDP), aggravated VCAM1 and ICAM1 expression suggesting an opposing effect of the diol to the epoxy lipid [71]. Furthermore, Ulu et al. demonstrated 19,20-EDP contributed to the antihypertensive actions of DHA in angiotensin II-induced hypertension. Importantly, the combination of 19,20-EDP with the sEHi TPPU (1-trifluoromethoxyphenyl-3-(1-propionylpiperidin-4-yl) urea) significantly lowered blood pressure as compared to angiotensin-II infused animals [72]. The current study was designed to first demonstrate the cardioprotective properties of AS-27 (19,20-EDP analogue), AA-4 and SA-26 (16,17-EDP analogues) as potential drug leads for IR injury and future experiments combining these compounds with sEHi are warranted. 

SIRT3, the primary mitochondrial deacetylase [13], has a key role in regulating mitochondrial function including regulating energy metabolism, ATP production, antioxidant defense and inflammatory signaling [16,18,73]. SIRT3 directly deacetylates and activates the mitochondrial antioxidant MnSOD significantly enhancing its ability to scavenge ROS [18,49,50]. SIRT3 promotes effective electron transport via the deacetylation of ETC complex components, the main source of more than 90% of cellular ROS, which indirectly reduces ROS production [48]. Experimental studies demonstrate that cardiac SIRT3 levels and/or activity are decreased in response to acute IR injury [24]. This is supported by evidence in H9c2 cells or adult mouse hearts where knocking down SIRT3 aggravated IR injury and mitochondrial dysfunction [23]. Furthermore, ex vivo studies using hearts isolated from SIRT3^-/-^ mice subjected to global IR exhibited significantly less recovery of cardiac function [74]. Approaches to upregulate SIRT3 expression and activity, such as administration of melatonin prior to reperfusion resulted in decreased acetylation of MnSOD, improved postischemic cardiac contractile function and limited oxidative damage [75]. In agreement with these reports, the current study demonstrated that hearts subjected to IR injury had decreased SIRT3 activity, increased acetylation of MnSOD, reduced ATP content and accumulation of MDA. Interestingly, the perfusion of hearts with the EDP analogue SA-26 during the reperfusion period was able to maintain SIRT3 activity, reduced acetylation of MnSOD and ameliorated the accumulation of MDA which correlated with improved post-ischemic functional recovery. These effects were associated with improved cardiac ATP levels. Together, these data indicate maintenance and promotion of SIRT3 activity can protect cardiomyocytes from oxidative stress-mediated cell death and limit cardiac injury.

The rate-limiting factor of SIRT3 activity is the availability of NAD^+^, a required cofactor for functional deacetylase activity of the enzyme [53,54]. The observed reduction of SIRT3 activity in hearts subjected to IR injury could be attributed to the reduction of NAD^+^/NADH ratio. Di Lisa et al. reported that NAD^+^ depletion during IR injury is partly due to the opening of the mitochondrial permeability transition pore (MPTP) leading to a release of mitochondrial NAD^+^ into the cytosol where it is degraded by glycohydrolase [76,77]. Furthermore, during IR injury, ROS-induced damage leads to the production of single-strand breaks in the DNA which increases catabolism of NAD^+^, resulting in reduced level of NAD^+^ [78]. The reduction of NAD^+^ content limits the activation of SIRT3, thereby leading to a hyper-acetylation of MnSOD and thus further ROS production. This hypothesis is supported by the recent finding of Fu et al. demonstrating *trans*-viniferin, a natural derivative stilbenic antioxidant, can activate SIRT3 by increasing the NAD^+^/NADH ratio in a cell model of Huntington’s disease [79]. Consistent with this notion, our results revealed IR injury depleted NAD^+^ levels and consequently the NAD^+^/NADH ratio in the heart. However, hearts perfused with the analogue SA-26 maintained significantly higher NAD^+^/NADH ratio which could explain the improved SIRT3 activity and the enhanced post-ischemic cardiac recovery.

Proteins regulating mitochondrial dynamics, particularly DRP1 and OPA1, are involved in modulating mitochondria structure and function following IR injury [55]. Under healthy conditions, DRP1 is predominantly found in the cytosol as an inactive phosphorylated form. Following cellular stress, DRP1 shuttles to the outer mitochondrial membrane initiating fission events reducing the number of functional mitochondria accelerating myocardial cell death [57]. Moreover, several reports reveal limiting mitochondrial accumulation of DRP1 will protect cardiomyocytes against IR injury and improve cardiac function [80,81]. Conversely, OPA1 is localized in the inner mitochondrial membrane and has a critical role in maintaining cristae structure [58,59]. Damage from IR injury can induce OPA1 proteolysis which impairs mitochondrial function, perturbs cristae structure, induces fragmentation, thereby increasing susceptibility of cell death [59,60,82]. Data generated from the current study support these reports where we observed a marked increase in mitochondrial DRP1 expression and decrease in mitochondrial OPA1 levels associated with impaired cardiac functional recovery in hearts subjected to IR injury. However, the perfusion of hearts with SA-26 significantly limited this response, which supports the notion of reduced mitochondrial injury. However, neither AA-4 nor AS-27 prevented the mitochondrial localization of DRP1, which resulted in marked decreases of OPA1 levels.

The relationship between SIRT3 and proteins regulating mitochondrial dynamics is well-documented in other disease models, where the SIRT3-dependent inhibition of fission is accompanied by the maintenance and activation of fusion. For example, in a cisplatin-induced acute kidney injury model, the reduction of SIRT3 mRNA and protein expression is associated with a marked mitochondrial accumulation of DRP1, severe mitochondrial damage and severe injury than WT animals [83]. Hyperacetylated OPA1 exhibits reduced activity correlating with pathological hypertrophy observed in mouse hearts subjected to either angiotensin-II treatment or transverse aortic constriction. Enhanced activation or overexpression of SIRT3 protected these hearts by directly binding to and deacetylating OPA1 thereby augmenting its GTPase activity [20]. Together, these studies suggest SIRT3 is a crucial regulator of mitochondrial integrity and is essential to maintain fitness of mitochondrial population under stress conditions. 

Mounting evidence suggests that mitochondrial damage can trigger the activation of the NLRP3 inflammasome cascade which in turn aggravates mitochondrial dysfunction leading to a vicious cycle of continued injury and reduced cardiac function [29,30]. The assembly of the NLRP3 inflammasome requires recruitment of NLRP3 protein to mitochondria in the perinuclear region resulting in dissipation of membrane potential, decreased oxidative phosphorylation and mitochondrial damage exacerbating cardiac damage [63,64,84]. Interestingly, Chen et al. demonstrated SIRT3 overexpression significantly diminished NLRP3 inflammasome activation induced by trimethylamine *n*-oxide (TMAO) in endothelial cells [31]. Additionally, Liu et al. showed SIRT3-deficient macrophages display impaired autophagy associated with accelerated NLRP3 inflammasome activation and endothelial dysfunction [85]. Our data support these findings where increased NLRP3 and cleaved caspase-1 protein levels were observed in the mitochondrial fractions of hearts subjected to IR injury. However, hearts perfused with the analogue SA-26 showed significantly lower levels of NLRP3 and cleaved caspase-1 as well as caspase 1 activity consistent with the maintained SIRT3 activity and improved cardiac functional recovery. Neither AS-27 nor AA-4 significantly inhibited the activation and translocation of either NLRP3 or mature caspase-1 to the mitochondria which matched the impaired SIRT3 activity and the compromised cardiac recovery observed in these hearts.

Considering the numerous promising and salutary biological actions of EDPs, one goal of this study was to provide proof of concept that more stable, synthetic analogs of DHA epoxides could mimic the biological actions of the natural CYP metabolites. We developed EDP analogues because of the limited stability and storage conditions of endogenous EDPs. Our results demonstrate the EDP surrogate SA-26 protects against myocardial IR injury by limiting NLRP3 inflammasome activation, while other structurally similar compounds, AS-27 and AA-4, did not provide sufficient protection. Although the precise molecular mechanisms remain unknown, we propose that SA-26 mediates its cardioprotection by preserving SIRT3 function. This study provides new perspectives for the development of pharmacological agents based on the EDP structure and suggest SA-26 is a strong lead scaffold for future development of a clinical drug candidate. Importantly, the increased stability of SA-26 may serve as a potential therapeutic agent in limiting mitochondrial damage and myocardial injury in response to IR injury. Future studies are still required to assess the bioavailability of these analogues, and to identify how changing the parent EDP chemical structure via altering the number and/or the position of the double bonds as well as the inclusion of different substituted groups will affect the stability and potency of the generated compounds.

## 4. Materials and Methods 

### 4.1. Synthesis of EDP Surrogates

The EDP analogs were prepared in the Falck lab using standard synthetic methods as previously described [86,87]. All analogs were fully characterized by ^1^H/^13^C Nuclear Magnetic Resonance (NMR) and mass spectroscopy and were ≥95% pure. Details of synthesis of the compounds and ^1^H/^13^C NMR and mass spectra are given in the Appendix A.

### 4.2. Animals 

All studies were carried out using 2–3-month-old male and female wild-type (WT) C57/Bl6 mice weighing 25–30 g. Mice were maintained in a colony at the University of Alberta and housed under conditions of constant temperature and humidity with a 12:12-h light–dark cycle. Mice were fed on a standard rodent chow diet ad libitum (fat 11.3%, fiber 4.6%, protein 21% (w/w)). The composition of the diet includes linolenic acid (0.27%), linoleic acid (2.12%), arachidonic acid (0.01%), omega-3 fatty acid (0.45%), total SFA (0.78%) and total MSFA (0.96%) (PicoLab® Rodent Diet 20 Cat. No 5053, LabDiets, Inc., St. Louis, MO, USA). All animal experimental protocols were approved by the University of Alberta Health Sciences Welfare Committee (University of Alberta Animal Welfare, ACUC, study ID#AUP330) and conducted according to strict guidelines provided by the Guide to the Care and Use of Experimental Animals (Voloum. 1, 2nd ed., 1993, from the Canadian Council on Animal Care).

### 4.3. Isolated Heart Perfusion

Wild-type (WT) mice of both sexes (equal ratios) were anesthetized by an intraperitoneal injection of sodium pentobarbital (Euthanyl, 100 mg/kg). Following complete non-responsiveness to external stimulation, hearts were quickly excised and perfused in the Langendorff mode [40,88] with Krebs-Henseleit buffer containing (in mM) 120 NaCl, 25 NaHCO_3_, 10 Dextrose, 1.75 CaCl_2_, 1.2 MgSO_4_, 1.2 KH_2_PO_4_, 4.7 KCL, 2 Sodium Pyruvate (pH 7.4) and bubbled with 95% O_2_ and 5% CO_2_ at 37 °C. The left atrium was then excised, and a water-filled balloon made of saran plastic wrap was inserted into the left ventricle through the mitral valve. The balloon was connected to a pressure transducer for continuous measurement of LVDP and heart rate (HR). Hearts with persistent arrhythmias or LVDP less than 80 cm H_2_O were excluded from the experiment. Mouse hearts were perfused in the retrograde mode at a constant flow rate for 20 min of baseline (stabilization) and then subjected to 30 min of global no flow ischemia followed by 40 min of reperfusion. Hearts were perfused with either AS-27 (1 µM), SA-26 (1 µM), or AA-4 (1 µM). The concentrations utilized in the current study were based on previously published data from cell culture experiments and experience working with CYP-derived epoxy metabolites of *n*-3 PUFAs, which demonstrate cardioprotective effects at similar concentrations [89,90]. In all experiments, chemicals were added at the beginning of reperfusion and were present in the heart throughout the reperfusion period. The percentage of left ventricular developed pressure (%LVDP) at 40 min of reperfusion (R_40_), as compared to baseline LVDP, was taken as a marker for recovery of contractile function. After 40 min of reperfusion, hearts were immediately frozen and stored below −80 °C. Haemodynamic parameters were acquired and analyzed using ADI software from (Holliston, MA, USA). Collection of the heart effluent was taken during both pre- and postischemic protocols to determine of coronary flow (CF) rates.

### 4.4. Immunoblotting 

Frozen mouse hearts were ground, homogenized and then fractionated into mitochondrial and cytosolic fractions as previously described [41]. Briefly, frozen cardiac tissues were ground with mortar and pestle on dry ice and then homogenized in ice-cold homogenization buffer (20 mmol/L Tris–HCL, 50 mmol/L NaCl, 50 mmol/L NaF, 5 mmol/L sodium pyrophosphate, 1 mmol/L EDTA, and 250 mmol/L sucrose added on the day of the experiment, pH 7.0). Samples were first centrifuged at 800× *g* for 10 min at 4 °C to separate the cellular debris. The collected supernatant was then centrifuged at 10,000× *g* for 20 min. The pellet was resuspended in homogenization buffer to obtain a mitochondrial-enriched fraction. The supernatant was ultra-centrifuged at 105,000× *g* for 60 min and the subsequent supernatant was used as the cytosolic fraction. Protein concentrations in both cytosolic and mitochondrial fractions were measured by the Bradford assay. Western blotting was carried out as previously described [40]. Protein (30–50 μg) was resolved by electrophoresis on (10–15%) SDS-polyacrylamide gels and transferred onto polyvinylidene difluoride (PVDF) membranes (BioRad Laboratories, Hercules, CA, USA). Immunoblots were probed with antibodies against CS (ab129095), caspase 1 (ab179515), total MnSOD (ab13533), acetyl-MnSOD (ab137037), VDAC (ab14734) (Abcam, Burlingame, CA, USA), SDH-A (5839), COX IV (11967), SIRT3 (5490), NLRP3 (15101), DRP1 (5391) (Cell Signaling Technology, Inc., MA, USA) and OPA1 (612606) (Becton Dickinson Canada Inc, Mississauga, ON, CAN). After washing, membranes were incubated with the corresponding secondary antibodies. The blots were visualized with ECL reagent. Relative band intensities were expressed as fold of the control assessed using Image J software (NIH, USA).

### 4.5. Measurement of MDA Levels

The level of MDA was assessed in the cardiac tissue using a lipid peroxidation (MDA) colorimetric assay kit (ab118970, Abcam, Burlingame, CA, USA) according to manufacturer’s instructions [41,52]. In this assay, free MDA present in the sample reacts with thiobarbituric acid (TBA) and generate a MDA-TBA adduct which was quantified colorimetrically at wavelength 532 nm. MDA levels were expressed as nmole MDA per mg protein.

### 4.6. Enzymatic Assays

Cleavage of the caspase-1 specific fluorogenic substrate Ac-YVAD-AMC (ALX-260-024-M005, Enzo life Sciences, NY, USA) was used to assess functional caspase-1 activity in cytosolic fractions of the heart homogenates [41]. The assay quantitated the fluorescence intensity of the cleaved 7-Amino-4-methylcoumarin (AMC) using a fluorometer (at excitation 380 nm, and emission 460 nm wavelengths). The activity was calculated by using a linear standard curve created with AMC and normalized to the sample protein concentration.

SIRT3 activity was detected in the isolated mitochondrial fractions using a SIRT3 fluorescent assay kit (50088, BPS Bioscience, San Diego, CA, United States), according to the manufacturer’s instructions [91]. In this assay, heart mitochondrial fractions were mixed with the specific HDAC fluorogenic substrate, bovine serum albumin, NAD^+^ and assay buffer. The deacetylation process induced by SIRT3 in the sample sensitizes the HDAC substrate so that subsequent treatment with the SIRT3 assay developer produces a fluorescence product that was measured using a fluorescence plate reader at 350/460 nm excitation/emission wavelengths. The activity of SIRT3 was expressed as U/μg protein.

### 4.7. Measurement of ATP Levels in the Heart

ATP levels in the heart tissue were measured using a fluorometric based assay kit (ab83355, Abcam Inc, Toronto, ON, Canada). Briefly, heart powders were homogenized in ice-cold assay buffer and centrifuged at 15,000× *g* for 2 min. The supernatant was used to quantitate ATP at 535 nm excitation and 587 nm emission with a Biotek plate reader (Winooski, VT, USA).

### 4.8. Measurement of NAD^+^/NADH Content in the Heart 

NAD^+^ and NADH contents were measured in whole heart homogenates using a colorimetric NAD/NADH Assay Kit according to the manufacturer’s protocol (ab65348, Abcam, Burlingame, CA, USA). Samples were split into 2 fractions to separately measure total amount of nicotinamide adenine dinucleotides (NAD^+^ + NADH = NADtotal) in one fraction and NADH in the other fraction after decomposition of NAD^+^ by heating the samples at 60 °C for 30 min. Levels of NAD^+^ were estimated by subtracting NADH from NADtotal. Absorbance was measured at 450 nm using microplate reader.

### 4.9. Statistics

Values are expressed as mean ± standard error of mean (SEM). Statistical significance was determined by one-way ANOVA with a Tukey post hoc test to assess differences between groups; *p* < 0.05 was considered statistically significant.

## Figures and Tables

**Figure 1 ijms-21-05261-f001:**
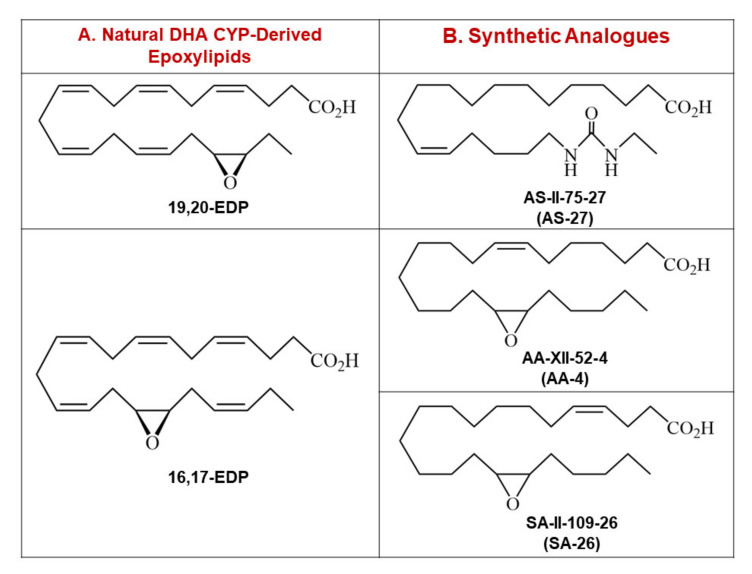
(**A**) Chemical structures of the two most abundant natural metabolites of DHA, 19,20-EDP and 16,17-EDP. (**B**) Chemical structures of AS-27, the synthetic analogue of 19,20-EDP, as well as AA-4 and SA-26, synthetic analogues of 16,17-EDP. DHA: Docosahexaenoic acid, EDP: Epoxydocosapentaenoic acid.

**Figure 2 ijms-21-05261-f002:**
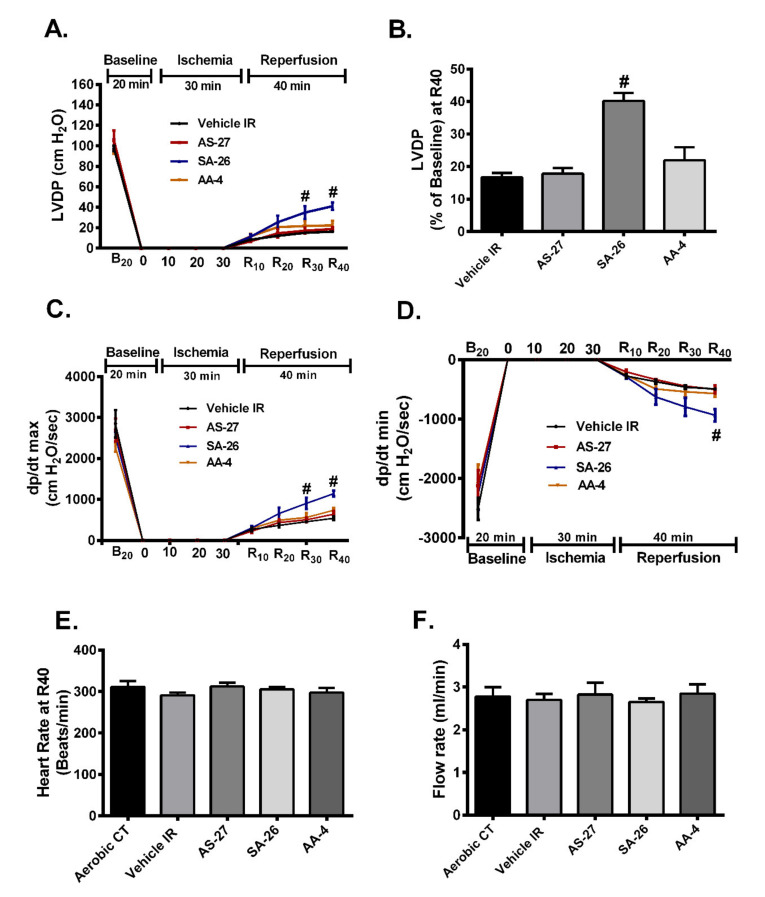
Perfusion of hearts with SA-26 improved postischemic contractile parameters. (**A**) LVDP at the baseline before drug treatment (B_20_), during ischemia, and at 10, 20, 30 and 40 min reperfusion (R_10_, R_20_, R_30_, and R_40_). (**B**) LVDP recovery at 40 min of reperfusion as a percentage of baseline. (**C**) Rate of contraction (dP/dt max), and (**D**) rate of relaxation (dP/dt min) at baseline before drug treatment (B_20_), ischemia, and 10, 20, 30 and 40 min reperfusion (R_10_, R_20_, R_30_, and R_40_). (**E**) Heart rate assessed as beats per minute (BPM) in vehicle treated aerobic hearts and hearts subjected to ischemia-reperfusion (IR) injury at the end of reperfusion (R_40_). (**F**) Coronary flow rates from vehicle treated aerobic hearts as well as hearts subjected to IR injury assessed post-ischemia. Values represent mean ± SEM; # *p* < 0.05 vs. Vehicle IR (*n* = 4–7 per group). LVDP; left ventricular developed pressure.

**Figure 3 ijms-21-05261-f003:**
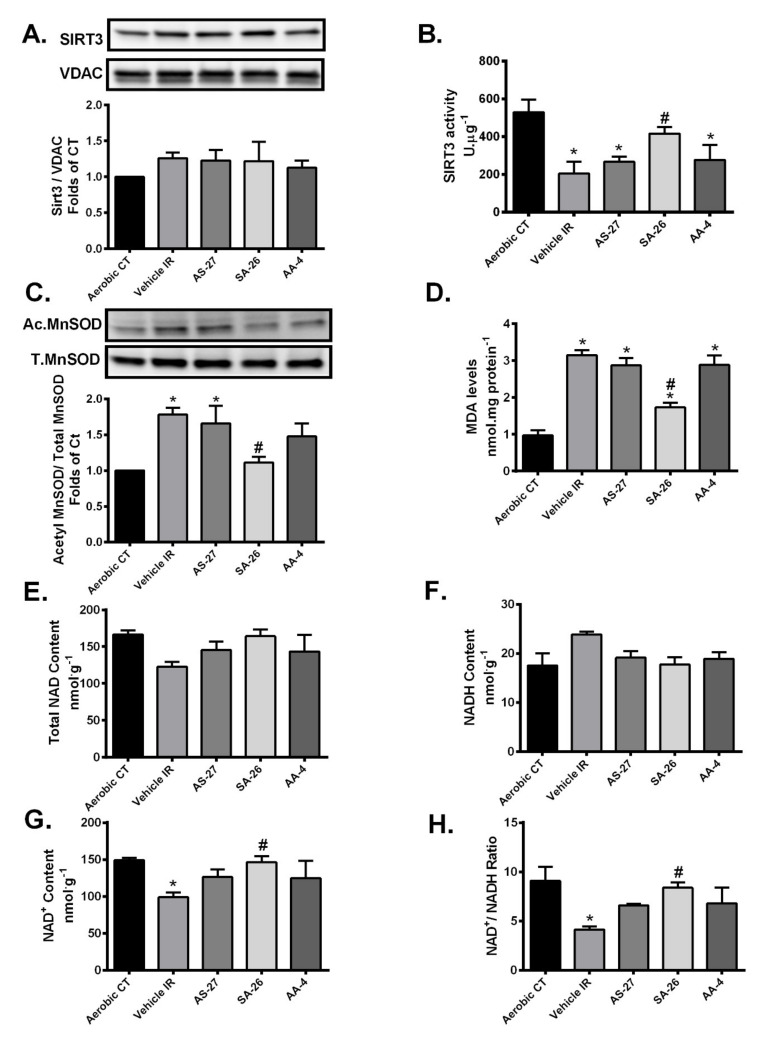
Preservation of NAD^+^/NADH control ratio and SIRT3 activity limited oxidative stress following IR injury. (**A**) Representative immunoblots and densiometric quantification of the expression of mitochondrial protein SIRT3 in mice hearts after 30 min ischemia and 40 min reperfusion. Protein expression was normalized to voltage-dependent anion channel (VDAC) protein used as a loading control. (**B**) Cardiac SIRT3 activity was determined in mitochondrial fractions in mice hearts after 30 min ischemia and 40 min reperfusion using a SIRT3 fluorescent assay kit. (**C**) Representative immunoblots and densiometric quantification of the relative protein expression of AcMnSOD normalized to total manganese superoxide dismutase (MnSOD) in mice hearts after 30 min ischemia and 40 min reperfusion. (**D**) Cardiac malondialdehyde (MDA) levels assessed using a lipid peroxidation (MDA) colorimetric assay kit in mice hearts following 30 min ischemia and 40 min reperfusion. Cardiac levels of (**E**) Total NAD, (**F**) NADH, (**G**) NAD^+^ and (**H**) NAD^+^/NADH ratio expressed as nmol per g dry weight assessed in mice hearts after 30 min ischemia and 40 min reperfusion. Values represent mean ± SEM, * *p* < 0.05 vs. Aerobic CT (control), # *p* < 0.05 vs. Vehicle IR (*n* = 4–5 per group).

**Figure 4 ijms-21-05261-f004:**
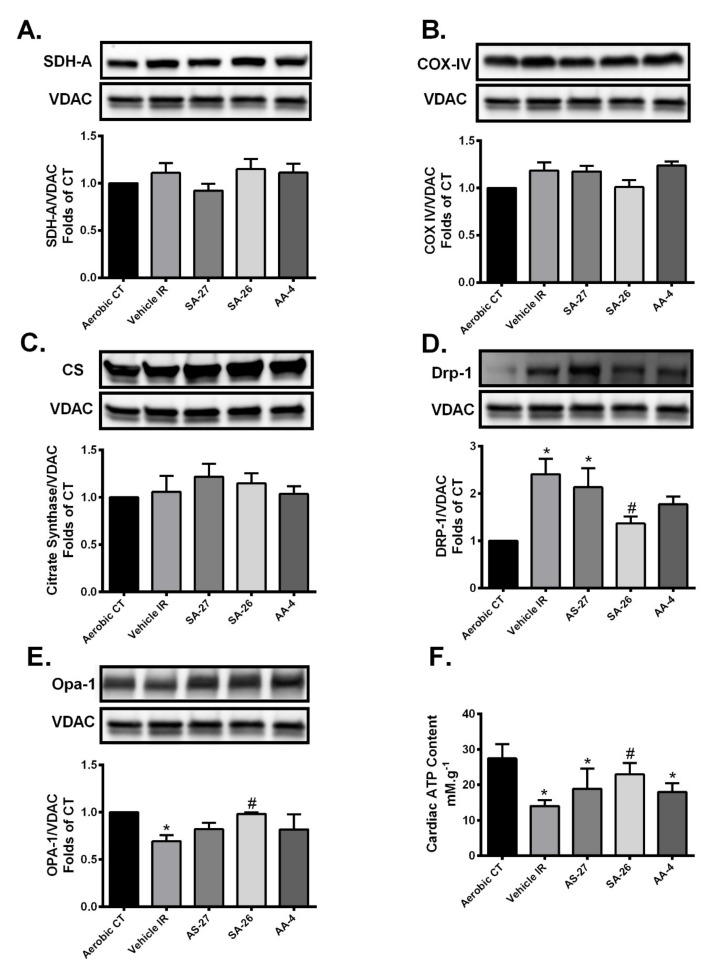
Perfusion of hearts with SA-26 preserved mitochondrial integrity and function under IR insult. Representative immunoblots and densiometric quantification of the mitochondrial protein expression of (**A**) succinate dehydrogenase subunit A (SDH-A), (**B**) cytochrome c oxidase subunit IV (COX IV), (**C**) citrate synthase (CS), (**D**) dynamin-related protein 1 (DRP1) and **(E**) optic atrophy 1 (OPA1) proteins observed in mice hearts following 30 min ischemia and 40 min reperfusion. All expressions were normalized to VDAC loading control. (**F**) Cardiac ATP levels were measured in mice hearts using a fluorometric based assay kit assessed following 30 min ischemia and 40 min reperfusion. Values represent mean ± SEM, * *p* < 0.05 vs. Aerobic CT, # *p* < 0.05 vs. Vehicle IR (*n* = 4–5 per group).

**Figure 5 ijms-21-05261-f005:**
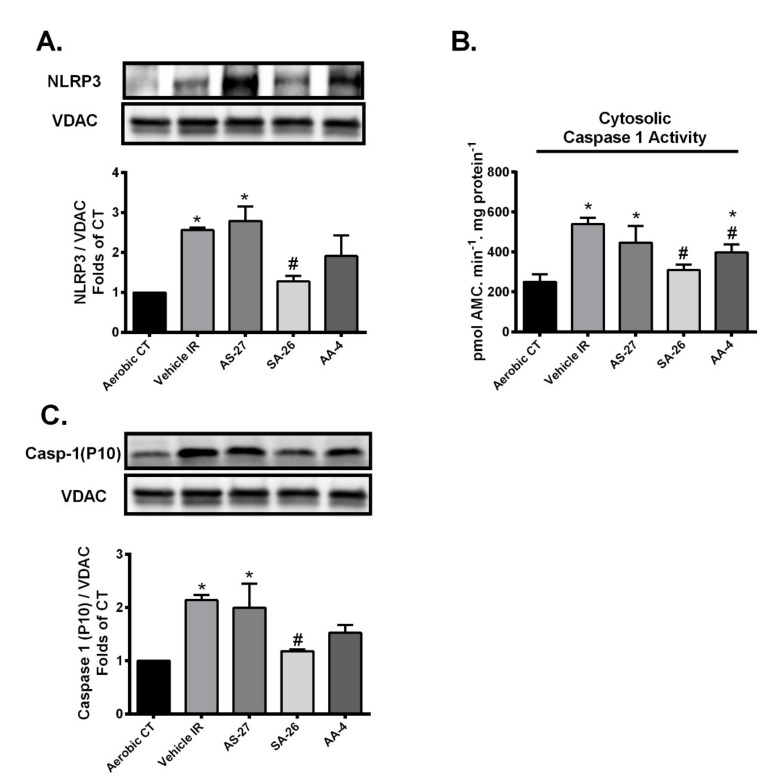
Perfusion of hearts with SA-26 inhibited the IR-induced activation and translocation of the NLRP3 inflammasome to the mitochondria. (**A**) Representative immunoblots and densiometric quantification of the mitochondrial expression of NLRP3 protein in mice hearts after 30 min ischemia and 40 min reperfusion. Protein expression was normalized to VDAC loading control. (**B**) Cardiac caspase-1 enzymatic activity assessed in the cytosolic fraction following 30 min ischemia and 40 min reperfusion. The assay quantitated the fluorescence intensity resulting from the cleavage of the caspase-1 specific fluorogenic substrate Ac-YVAD-AMC by the cytosolic heart homogenates. (**C**) Representative immunoblots and densiometric quantification of the mitochondrial expression of cleaved caspase-1 (P10) in mice hearts after 30 min ischemia and 40 min reperfusion. Protein expression was normalized to VDAC protein used as a loading control. Values represent mean ± SEM, * *p* < 0.05 vs. Aerobic CT, # *p* < 0.05 vs. Vehicle IR (*n* = 4 –5 per group).

**Figure 6 ijms-21-05261-f006:**
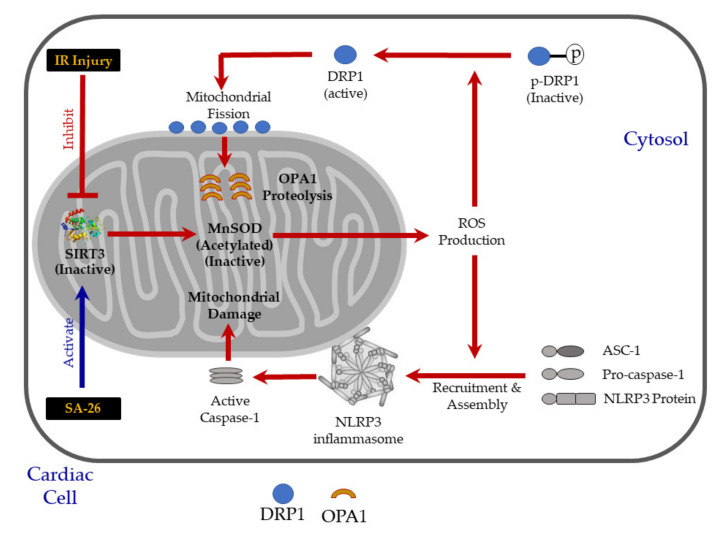
Conceptual illustration showing the potential cardioprotective role of the EDP surrogate SA-26 against IR injury. Hearts subjected to IR injury showed decreased SIRT3 activity and increased acetylation of MnSOD, suggesting diminished antioxidant capacity. Accordingly, the reactive oxygen species (ROS) burst from damaged mitochondria triggers excessive activation and recruitment of DRP1 on the mitochondrial outer membrane, inducing OPA1 proteolysis which impairs mitochondrial fusion, perturbs cristae structure, and prompts mitochondrial fragmentation. Furthermore, the excessive ROS production from the damaged mitochondria activates the assembly of NLRP3 inflammasome on mitochondrial membrane leading to the activation of caspase-1 which in turn, aggravates mitochondrial damage. Perfusion of hearts with SA-26 during the reperfusion period ameliorates IR injury by maintaining mitochondrial integrity and function. Hearts perfused with SA-26 showed improved SIRT3 activity, enhanced antioxidant capacity and limited oxidative injury. This effect was associated with the inhibition of the assembly and mitochondrial translocation of the detrimental NLRP3 inflammasome and consequently improved postischemic functional recovery.

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
