# Peer review of "A Synthetic Epoxydocosapentaenoic Acid Analogue Ameliorates Cardiac Ischemia/Reperfusion Injury: The Involvement of the Sirtuin 3–NLRP3 Pathway"

_ijms, 2020, doi:10.3390/ijms21155261_

Round 1
Reviewer 1 Report
In this manuscript the authors have demonstrated the cardioprotective effects of the EDP analogue SA-26 against IR injury that involve preservation of mitochondrial SIRT3 activity. The protective effect of SA-26 compared to AA-4 and Sa-27 is evident. SA-26 has a protective effect which, however, is limited because in the different experiments, the values observed in the presence of SA-26 do not correspond to control. However, the SA-26 protection is interesting and worthy of being deepened.
Minor revision
In this manuscript the authors have demonstrated the cardioprotective effects of the EDP analogue SA-26 against IR injury. SA-26 is able to preserve mitochondrial SIRT3 activity and inhibit NLRP3. The protective effect of SA-26 compared to other evaluated EDP analogues AA-4 and Sa-27 is evident. The topic is interesting but there are some criticisms:
- In Fig. 2E and 2F Aerobic CT would be necessary to understand if heart rate and flow rate are in any case influenced with respect to a condition not injured.
- In Fig. 3B- 3C- 3D, the effect of SA-26 is significant compared to vehicle IR, but SA-26 values are always lower than the aerobic CT. The SA-26 cardioprotection is partial and other factors could contribute. Furthermore, could this partial protection have consequences?
- In paragraph 2.3 the authors analyze the protein levels of ETC components SDH-A and COX IV in different experimental conditions and no differences are observed. It would be more complete and accurate to analyze all the ETC components.
- Citrate synthase is an enzyme active that catalyze the first reaction of the Krebs Cycle. Why did the authors analyze the level of this protein? Please explain the rationale
- In paragraph 2.3 the authors state that they observe a marked increase in mitochondrial expression of DRP1 and a significant decrease in mitochondrial OPA1 levels in the vehicle IR group compared to the aerobic CT (Fig. 4D/E). Western blot images confirm DRP1 increase but not OPA1 decrease. Furthermore OPA-1 Western blot image do not correspond that is shown in the graph for OPA1. Why? How many times has this experiment been performed?
- In Cardiac ATP content, ATP value for SA-26 is lower than aerobic CT. Could this reduction of ATP content cause problems for a full recovery?
- Fig.5A western blot do not correspond to value reported in graph. Differences in NLRP3 level between Vehicle IR and Sa-26 are not so evident
- What is the direct target of SA-26?
Reviewer 2 Report
Ischemia-reperfusion (IR) injury is the primary pathological manifestation of ischemic heart disease, which can lead to permanent disability or event to death. Thus, the identification of compounds and strategies that protect the myocardium from IR injury is of high importance. In this manuscript, Ahmed M. Darwesh and coworkers describe the cardioprotective properties of epoxydocosapentaenoic acid (EDPs) derivatives. Specifically, the authors show that one the EDPs derivatives, SA-26 protects against myocardial IR injury by decreasing NLRP3 inflammasome activation. Interestingly, they also found that other related compounds AS-27 and AA-4 do not have such effect/s. Overall, the data presented in the manuscript is very interesting for the field and of high quality. This reviewer has no important concerns and only some minor comments.
Minor comments:
- Have the authors tested the effects of different concentrations of the compound AS-27, SA-26 and AA-4? There is any dose-response effect for the compound SA-26 during heart perfusion? The authors mentioned in MM section that they evaluated a single concentration based on information already published data from CYP-derived epoxy metabolites in cellular models. However, it would be interesting to hear their opinion on this
- Related to my previous comment, have the authors evaluated the effects of the compounds AS-27, SA-26 and AA-4 using cellular models? Do they have any evidence for the direct binding and activation of SIRT3 activity by the compound SA-26? Please explain this.
- Do the authors think that SA-26 constitutes a good starting point for further drug development? What are the strategies that the authors propose for further improvement? How do the authors explain the differences in the action between the three different compounds? It is possible that such differences can arise from differences in their chemical stabilities and/or metabolism? This should be further discussed.
Reviewer 3 Report
Comment to Authors:
This manuscript by Darwesh et al., discusses how synthetic EDP analogues of the 19,20-EDP and 16,17-EDP variety may be cardioprotective against ischemia reperfusion (IR) injury. The authors present dta that support these findings utilizing well-established models of IR injury, and conducting well-designed follow up experiments to assess the potential mechanisms by which this is occurring. The authors also delineate the mechanism by conducting certain assays (such as NAD/NADH or ATP measurements). Overall, the manuscript is leads way to future studies that can show how these analogues may be cardioprotective. However, there is a lack of rationale of why 16,17-EDP analogs were chosen for the study, why were some double bonds retained while others were removed and the rationale of why certain analogs worked better than the others are not explained clearly. Overall this is a very important and well done studies but lacks rationale in some sections.
Major Comments:
- It is stated in Figure 1 that 16,17- and 19,20-EDP are most abundant metabolites of DHA. Is this the primary reason for why analogues of these metabolites were synthesized? Are these the most important metabolites with respect to cardiovascular effects? How do these compare with the AA and EPA derivatives that were published before.
- What is the logic for the chemical changes made to the analogues? For instance, the manuscript states that a partially saturated carbon backbone avoids autooxidation and improves physical stability. It seems the rationale for the design is based on several previous paper from Falck group on the AA and EPA derivatives. It will be good to include it here. The previous excellent papers from Falck group could be recalled here do rationale for the design of the analogs.
- It would have been good to compare the analogs with 19,20-EDP and sEH inhibitor. But this will be hard experiment to add at this point. However, the authors can comment if there are any similar studies reported on 19,20-EDP + sEH in a cardiovascular model.
- The analogues being made of each EDP metabolite are drastically different. Do the authors have any studies using AA-4 or SA-26 with an epoxide at the 19,20- position as opposed to the 16,17-? While the studies are comparing different analogues, the analogues between the different metabolites are not consistent. Thus, it does not appear to be an appropriate comparison without justification for why those specific chemical modifications were made. Additionally, do the authors have a 16,17- synthetic analogue containing the urea like that in AS-27? Previously it was shown by Falck group the urea group is a good bioisotere of the eppoxide. The SA-26 urea derivative will be a good compound to test (but hard to make)
- Bioavailability of the EDP’s are mentioned twice in the manuscript prior to the statement that 16,17-EDP and 19,20-EDP analogues are synthesized. 1) While the experiments in this manuscript do not directly assess the bioavailability of these analogues, it would be greatly helpful to include a statement on this. 2) Are the decisions for the chemical structures of the analogues also considering that it will increase bioavailability? If yes then maybe a study can be done with the drugs in buffer or with blood.
- Please include the full methods for how these molecules were synthesized in addition to the reference to Adebesin et al., 2019. The reference 82 refers to EEQ analog. It will good to include all the details of the synthesis, NMR and characterizations here for the reader either in main text or supplementary.
Minor Comments:
- Page 2, Line 54-55. Statement is not clear. Do the activated NLRP3 inflammasomes have damaged mitochondria, and as a result release ROS? Do the damaged mitochondria release ROS, which activate the NLRP3 inflammasomes?
- Page 3, Line 92-93: Do the authors intend to state that the current study investigates the properties of more chemically and metabolically stable synthetic analogues?
